# SMYRF:
# Efficient Attention using Asymmetric Clustering

**Giannis Daras**
Computer Science Department
The University of Texas at Austin
giannisdaras@utexas.edu

**Augustus Odena**
Google Research
augustusodena@google.com

**Nikita Kitaev**
Google Research
kitaev@cs.berkeley.edu

**Alexandros G. Dimakis**
ECE Department
The University of Texas at Austin
dimakis@austin.utexas.edu

## Abstract

We propose a novel type of balanced clustering algorithm to approximate attention. Attention complexity is reduced from $O(N^2)$ to $O(N \log N)$, where $N$ is the sequence length. Our algorithm, SMYRF, uses Locality Sensitive Hashing (LSH) in a novel way by defining new Asymmetric transformations and an adaptive scheme that produces balanced clusters. The biggest advantage of SMYRF is that it can be used as a drop-in replacement for dense attention layers *without any retraining*. On the contrary, prior fast attention methods impose constraints (e.g. queries and keys share the same vector representations) and require re-training from scratch. We apply our method to pre-trained state-of-the-art Natural Language Processing and Computer Vision models and we report significant memory and speed benefits. Notably, SMYRF-BERT outperforms (slightly) BERT on GLUE, while using $50\%$ less memory. We also show that SMYRF can be used interchangeably with dense attention before and after training. Finally, we use SMYRF to train GANs with attention in high resolutions. Using a single TPU, we were able to scale attention to 128x128=16k and 256x256=65k tokens on BigGAN on CelebA-HQ.

## 1  Introduction

Attention layers enable long-range representation learning and are becoming indispensable in architectures for both Image Synthesis [1, 2, 3] and Natural Language Processing [4, 5, 6, 7, 8, 9]. Attention finds further uses in other domains like symbolic mathematics and music modeling as well [10, 11, 12]. Unfortunately, attention layers have high computational and memory cost which scales quadratically in the size of the input sequence. This constraint is so onerous that the canonical implementation of attention for image synthesis - Self-Attention GAN [2] - could only afford to use one self-attention layer. For NLP, modern transformer-based models can only be trained in large industry research labs with massive infrastructure investments. For instance, the recently published GPT-3 [13] model uses 96 attention layers trained on input sequences of 2048 tokens. When fine-tuning pre-trained attention models, NLP researchers usually truncate input sentences, limiting performance on datasets with longer inputs.

Recent research [14, 3] indicates that dense attention is statistically and computationally inefficient [15, 16, 3]: it does not account for the locality inherent in many tasks. Alternatives have been proposed that are either more efficient [12, 17, 18, 19, 20, 7, 21, 22] or that better accommodate locality [23, 3]. Most such alternatives have been sparse. Sparsity can be achieved by limiting

attention to pre-defined positions [23, 3, 22, 12]. Recent work [17, 18, 19, 20] proposes data-driven sparsity, which allows for discovery of arbitrarily complex dependencies between input positions.

Despite this progress, new state-of-the-art models [8, 13, 9, 24, 25, 26] still use the original dense attention layers. There are three reasons for this: (i) alternative fast-attention mechanisms degrade the performance of the underlying model. For example, replacing dense attention layers in Transformers with memory efficient local attention [23] increases perplexity from $41.57$ to $44.23$ [20]. (ii) some mechanisms work well, but make very strict assumptions. For example, in Star Transformer [22] all nodes attend to a relay node which summarizes the content of the entire input sequence, but this prevents the use of causal masking, so it can only be used for encoding. (iii) some alternatives are only efficient in theory. For example, in some variants [17, 27] sparsification of the attention map happens after instantiating the matrix, and so quadratic memory is still used before instantiation. Finally, [12, 28] require highly specialized GPU-kernels and which prevents usage in several hardware settings (e.g. TPUs). The design of fast and efficient attention layers remains a challenge.

**Our Contributions:**
**1)** We propose a novel type of balanced clustering to approximate attention. We call the underlying optimization problem Attention Biclustering and prove that finding an exact solution is computationally intractable.
**2)** We propose an algorithm for solving Attention Biclustering efficiently in practice. Our algorithm, SMYRF, uses Locality Sensitive Hashing (LSH) in a novel way by defining new Asymmetric transformations and an adaptive scheme that produces balanced clusters.
**3)** Our method, SMYRF, can handle different query and key vectors, just like normal dense attention. As a result, SMYRF layers are drop-in replacements for pre-trained models, unlike previously proposed fast-attention mechanisms such as Sinkhorn [20], Reformer [18] and Routing Transformer [19].
**4)** We show through numerous experiments that SMYRF attention layers are very effective in terms of performance, memory and speed, even without any training. We measure the memory-performance trade-off of applying SMYRF to state-of-the-art NLP and Computer Vision models, across more than a dozen tasks. For example, we are able to shrink the memory requirements of a pre-trained BigGAN [1] by $50\%$ while maintaining $98.2\%$ of its Inception score without re-training.
**5)** We finetune SMYRF on GLUE [25] starting from a BERT (base) checkpoint. We demonstrate that SMYRF-BERT outperforms BERT while using $50\%$ less memory. We also show that with $75\%$ less memory, SMYRF maintains $99\%$ of BERT performance on GLUE. Due to SMYRF's portability, we are also able to conduct experiments for various memory configurations with pre-trained BERT and RoBERTa [9] models on IMDB. We show slight performance drops for great memory benefits.
**6)** We show that SMYRF can be interchanged with dense layers *before* and *after* training. We report performance gains by using SMYRF in a back-and-forth manner: we replace dense with SMYRF during training (to earn in memory) and we replace SMYRF with dense attention during inference (to earn in performance). The interchangeability of SMYRF with dense attention is unique, as it has not been observed in previously proposed attention alternatives [18, 19, 20, 28, 3].
**7)** We are able to scale the resolution of attention for GANs, due to our reduced memory footprint. We train a BigGAN with an $128 \times 128$ SMYRF attention layer and show it outperforms the dense attention performance, decreasing FID from $26.06$ to $25.03$ in Celeba-HQ-128 [29]. Finally, we successfully train a BigGAN with attention at resolution $256 \times 256$ on a single v3-8 TPU.
**8)** We open-source our code and pre-trained models to encourage more related research: https://github.com/giannisdaras/smyrf.

## 2  Background

Attention [30] works by computing inner products of query and key vectors. Depending on the application, these vectors may represent embeddings for tokens or image pixels. Input of each attention layer is three sets: $\mathcal{Q}, \mathcal{K}, \mathcal{V}$ for query, key and value vectors respectively. Attention of $q$ to the keys set $\mathcal{K}$ outputs a new vector $o_q$, which is a weighted sum of value vectors $v_i \in \mathcal{V}$ where each weight $w_i$ increases with the inner product $q \cdot k_i$. Specifically, the output is computed as:

$$o_q = \sum_{i=1}^{N} w_i v_i, \qquad w_i = \frac{e^{q \cdot k_i}}{\sum_{j=1}^{N} e^{q \cdot k_j}}. \tag{1}$$

Here, we assumed for notational simplicity that $N = |\mathcal{Q}| = |\mathcal{K}|$. Using matrix notation, attention is equivalently defined as $\sigma(Q \cdot K^T) \cdot V$ where $Q, K, V$ are matrices with rows the embeddings for each query, key, value and the function $\sigma(.)$ computes the row-wise softmax.

# 3 Approximating Attention with Clustering

## 3.1 Motivation

Our method is motivated by the observation that attention matrices have interesting structure in real datasets. Naively, to compute dense attention, as equation 1 shows, we need to compute all outputs $o_{q_i}$, i.e. $O(|\mathcal{Q}| \cdot |\mathcal{K}|)$, a quadratic number of inner products $q_i \cdot k_j$, $q_i \in \mathcal{Q}$, $k_j \in \mathcal{K}$. However, we observe that in most real networks, the attention weights $w_i$ are sparse, because of the softmax operation and the structure of the vectors. For example we observe that in a pre-trained BigGAN on ImageNet, on average $\mathbf{98.11 \pm 0.26\%}$[1] of keys get weight less than 0.01 in softmax and $\mathbf{86.11 \pm 2.92\%}$ of them get less than $\frac{1}{|\mathcal{K}|}$, where $\mathcal{K}$ is the number of keys.

Further, we observe that the attention matrix is near low-rank, even after the softmax. By definition, the matrix $Q \cdot K^T$ is going to be of rank at most the dimension of the query and key vectors. Therefore, if the embeddings dimension is smaller than the input sequence, the attention matrix is low-rank. This is more pronounced for images and long-context language models. However, one can easily construct cases of low-rank matrices which become full rank after softmax. Our finding is that this does not happen in practice. In the Appendix we show that *real attention matrices of pretrained models have a sharp decay in their singular values and hence can be well approximated by low-rank matrices*.

SMYRF benefits from sparsity and low-rank structure of attention matrices. By clustering keys and queries into groups, we obtain block-diagonal structure in the approximate attention matrix, since only query-key pairs within the same cluster are computed. We show that this method leads to accurate approximations of dense attention and it can be computed much faster and with much less memory.

## 3.2 Problem Formulation

We formulate the assignment of keys and queries into clusters as an optimization problem. Denote with $P_{ij} = q_i^T k_j$ the element $(i, j)$ of the product matrix $P = Q \cdot K^T$ and the attention map with $M = \sigma(Q \cdot K^T)$. We will assign query and key vectors into $L$ clusters $c_1, c_2, ..., c_L$ and compute attention only within each cluster. For fast execution on TPUs/GPUs, all partial attentions should be computed in parallel. For this reason, we require that clusters are balanced: i.e. all clusters contain the same number of keys and queries. We note that the number of keys in each cluster does not have to be equal to the number of queries. Formally, each cluster contains $\frac{|\mathcal{Q}|}{L}$ queries and $\frac{|\mathcal{K}|}{L}$ keys.

We denote with $\mathcal{C}^L$ the set of all possible assignments in $L$ balanced non-overlapping clusters. A specific assignment is denoted by $\mathcal{C}_t^L$ and there are $T$ possible such assignments, where $T$ is exponentially large in the number of keys and queries.

$$\mathcal{C}^L = \{\mathcal{C}_1^L, \mathcal{C}_2^L, ... \mathcal{C}_T^L\}.$$

$$\mathcal{C}_t^L = \{c_1, c_2, ..., c_L\} : \quad \begin{cases} c_i = \{q_1, ..., q_{\frac{|\mathcal{Q}|}{L}}, k_1, ..., k_{\frac{|\mathcal{K}|}{L}}\}, & c_i \subseteq \mathcal{Q} \cup \mathcal{K}, \ \forall i \in \{1, ..., L\} \\ c_x \cap c_y = \varnothing \quad \forall c_x, c_y \in \mathcal{C}_t^L. \end{cases}$$

$$(2)$$

We emphasize that every key and query is assigned in a unique cluster for any valid assignment $\mathcal{C}_t^L$: $c_x \cap c_y = \varnothing \quad \forall c_x, c_y \in \mathcal{C}_t^L$. We also define a masking operator $\text{Mask}_\epsilon$ that takes as input: (i) a clustering $\mathcal{C}_t^L \in \mathcal{C}^L$ and (ii) the product matrix $P$ and replaces $(q, k)$ pairs that are not in the same cluster with $-a$, where $a \in \mathbb{R}^+$ is a constant chosen to satisfy $e^{-a} = \epsilon$ for a given $\epsilon \geq 0$. Formally:

$$\text{Mask}_\epsilon(\mathcal{C}_t^L, P_{ij}) = \begin{cases} P_{ij} & \text{iff } \exists t : (i, j) \in c_t, \\ -a, & \text{o/w.} \end{cases}$$

Intuitively, the masking operator replaces inner products of queries and keys that are not in the same cluster with an arbitrarily small number, so that the softmax will assign a score arbitrarily close to zero to these entries. We denote with $\hat{P}_\epsilon = \text{Mask}_\epsilon(\mathcal{C}_t^L, P)$ the product matrix after the masking. With this notation, $\hat{P}_0 = \text{Mask}_0(\mathcal{C}_t^L, P)$, is the product matrix for the within-clusters attention.

**Attention Biclustering:** Under this formulation, we are searching for the cluster assignment $\mathcal{C}_t^L$ that approximates the dense attention matrix $\sigma(P)$ as well as possible, in Frobenius norm:

$$\min_{\mathcal{C}_t^L \in \mathcal{C}^L} ||\sigma(\hat{P}_0) - \sigma(P)||_F. \tag{3}$$

Note that $L$ must divide the number of queries and keys for this problem to be well-defined.

## 3.3 Complexity of Attention Biclustering

We start by showing that Attention Biclustering, the optimization problem defined in (3), is provably computationally intractable.

**Theorem 1.** *Attention Biclustering (3) is NP-hard.*

We defer the proof of this theorem to the Appendix. Our proof proceeds by first establishing hardness before the softmax, using a reduction from three dimensional matching [31]. We then leverage this to establish hardness of approximating attention through clustering after the softmax operation.

We consider it interesting to establish the computational intractability of Attention Biclustering, since this clustering formulation is quite unique due to the softmax operation. Our hardness result rules out an exact polynomial solution, unless P=NP. We propose an efficient algorithm that leverages hashing to assign queries and keys to clusters. Formally proving an approximation guarantee or provable inapproximability for the attention approximation problem we proposed remains open.

## 3.4 Proposed algorithm: SMYRF

Our algorithm consists of the following steps:
**1)** We first propose novel asymmetric transformations $F, G : \mathbb{R}^d \to \mathbb{R}^{d'}$ such that for all given queries $q_1, q_2 \in \mathcal{Q}$ and keys $k \in \mathcal{K}$: $q_1 \cdot k \le q_2 \cdot k \iff ||F(q_1) - G(k)||_2 \le ||F(q_2) - G(k)||_2$.
**2)** We then use a Locality Sensitive Hashing (LSH) function $h : \mathbb{R}^{d'} \to \mathbb{R}$ to map transformed vectors in real numbers, so that that vectors that are close in Euclidean distance correspond to numbers that are close on the real line.
**3)** We sort vectors based on their LSH value and group them by adapting the thresholds to ensure $L$ balanced clusters.
**4)** We perform dense attention within each cluster.

Our approximate attention algorithm relies on a few technical innovations:

**Novel Asymmetric Transformations:** We need an efficient way to find, for any given query vector $q_i \in \mathcal{Q}$ the set of keys with which it has big inner products. This problem, called Maximum Inner Product Search (MIPS), can be efficiently solved by transforming query and key vectors to convert it to a Nearest Neighbor Search (NNS) as proposed in the pioneering Asymmetric LSH (Locality Sensitive Hashing) work by Shrivastava et al. [32].

We are looking for functions $F : \mathbb{R}^d \to \mathbb{R}^{d'}, G : \mathbb{R}^d \to \mathbb{R}^{d'}$ such as: $||F(q) - G(k)||_2^2 = D(q \cdot k), \forall(q, k)$ where $D : \mathbb{R} \to \mathbb{R}$ a decreasing function that depends only on the inner product $q \cdot k$. We constrain our focus on functions $D$ that decrease linearly with the inner product $q \cdot k$. Several previous works have proposed Asymmetric LSH transformations [32, 33, 34] but focus on the case where we have a *single query* $q$ and multiple keys. In that case, any norm $||q||_a$ where $a = \{1, ..., \infty\}$ is constant and thus $D = D(q \cdot k, ||q||_a)$.

Our central algorithmic contribution is the proposal of novel asymmetric functions:

$$F(q_i) = \left[ q_i; 0; \sqrt{M_Q^2 + M_K^2 - ||q_i||_2^2} \right], \qquad G(k_i) = \left[ k_i; \sqrt{M_Q^2 + M_K^2 - ||k_i||_2^2}; 0 \right] \tag{4}$$

where we use the constants $M_Q = \max_{q_i} ||q_i||_2$, $M_K = \max_{k_i} ||k_i||_2$, or any other upper bound on the norms. With this transformation, all queries and keys are mapped to a $(d + 2)$-dimensional

ball with radius $\sqrt{M_Q^2 + M_K^2}$ and the distance of the transformed vectors decreases linearly with the inner product of the original vectors:

$$||F(q_i) - G(k_i)||_2^2 = 2 \cdot \left( M_Q^2 + M_K^2 - q_i \cdot k_i \right).$$ (5)

Note that the Euclidean distance of the transformed vectors depends only on the inner product of the original vectors and not on individual norms $||q_i||_2$ as in previous work [35, 34, 33]. We include details of comparison to the numerous prior asymmetric transformations in the Appendix.

**Adaptive Clustering:** The final step of SMYRF is to use the hashed values to create *balanced* clusters. These are created by forming balanced hash buckets where every group is assigned the same number of query and key vectors. We modify the E2LSH [35] hashes to create balanced clusters as follows: Instead of rounding the E2LSH to an integer value as in [35], we adaptively set the boundaries of the 1-d hashed space to ensure the same number of query and key vectors per interval. Computationally wise, this only requires sorting the hashes. We explain the mathematical details of our adaptive clustering scheme and the differences with E2LSH in the Appendix.

**Computational Complexity and speedups:** For notational simplicity we assume $|\mathcal{Q}| = |\mathcal{K}| = N$. The total time and memory complexity of SMYRF is $O\left( H \cdot N \cdot \log N + H \cdot \frac{N^2}{L} \right)$, where: $H$ denotes hashing rounds, $N$ number of query/key vectors and $L$ number of clusters. For most of our experiments we choose $L = O(N)$, $H = O(1)$, and thus complexity is $O(N \log N)$. Even though we obtain optimal complexity for $L = O(N)$, $H = O(1)$, both $L, H$ are parameters that can be tuned to satisfy the desired memory-performance trade-off. Regarding speed, SMYRF accelerates a lot attention as sequence length increases. For example, for sequence length 2048, SMYRF-BERT offers $\approx 20\%$ speedup, while for 4096 speedup increases to $\approx 50\%$. We include detailed speed plots for applying SMYRF to BERT in the Appendix.

## 4 Experiments

### 4.1 Pre-trained models

We first illustrate that SMYRF is an excellent drop-in replacement for pre-trained dense attention. We show significant memory benefits for relatively small performance drop, *with no training at all*. We use a pre-trained[2] BigGAN, which is a state-of-the-art model in Image Generation for ImageNet [37]. BigGAN has a single attention layer at resolution $64 \times 64$ (4096 queries). We replace BigGAN's dense attention with a SMYRF layer at the same resolution, with no other modifications. Figure 1 illustrates images generated by SMYRF-BigGAN for different memory savings, ranging from $99.44\%$ (first column) to $50\%$ (one to last column). Last column shows generated images using the dense attention layer ($100\%$ memory). As shown, SMYRF enables a new tradeoff in the design space. We can drastically reduce attention memory by $93.75\%$ with a small degradation or select any other point in this tradeoff depending on hardware specifications. We report a few Inception [38] and FID [39] scores for different memory savings in Table 1. We emphasize that no further modification was made to this model other than replacing the attention layer. By shrinking $50\%$ the memory requirements of attention, SMYRF maintains $98.2\%$ of Inception performance without any training. In the Appendix, we also include visualizations of clustering assignments in real-world images.

### 4.2 Finetuning pre-trained models

In this section, we *finetune* pre-trained models with SMYRF. We show that finetuned SMYRF models, with $50\%$ memory reduction, can outperform dense attention. We also show that even with more aggressive memory-shrinking, up to $97\%$, SMYRF maintains a relatively good performance.

We train SMYRF-BERT (base) on GLUE [25, 40, 41, 42, 43, 44, 45, 46, 47, 48, 49, 50, 51] benchmark, using sequence length 128. We compare the following five models: (i) BERT [6] (base), (ii) SMYRF-BERT (base) with $50\%$ memory reduction (2nd row), (iii) SMYRF-BERT (base) with $25\%$ memory reduction (3rd row), (iv) BERT (base) with input sequences truncated to 64 tokens ($50\%$ memory reduction, 4th row), (v) BERT (base) with input sequences truncated to 32 tokens

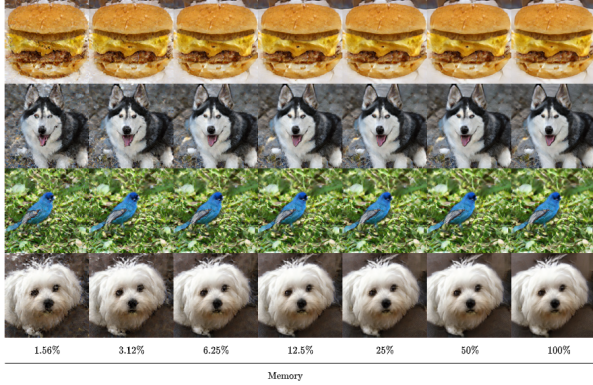

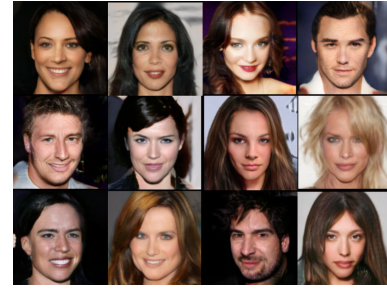

Figure 1: Images generated by SMYRF-BigGAN.
The model is initialized with the weights of a pre-trained BigGAN on ImageNet (no further training).
We show images for memory reduction ranging from 98.44% (first column) to 50% (one to last column).
Last column shows generated images by BigGAN with dense attention.

Figure 2: Generated images from SMYRF-BigGAN on Celeba-HQ-128. Attention at $128 \times 128$. The trained model uses 50% less memory compared to original BigGAN.

| | Memory | Rounds | $C$ | Inception | FID |
|---|---|---|---|---|---|
| BigGAN | 100% | 1 | 4096 | $\mathbf{93.79 \pm 1.96}$ | 11.30 |
| SMYRF-BigGAN | 50% | 32 | 64 | $91.91 \pm 2.65$ | 12.18 |
| | | 64 | 32 | $\mathbf{92.09 \pm 1.83}$ | 12.18 |
| | | 128 | 16 | $91.59 \pm 1.83$ | $\mathbf{12.10}$ |
| | 25% | 32 | 32 | $87.90 \pm 1.90$ | 13.34 |
| | | 64 | 16 | $88.45 \pm 1.70$ | 13.45 |
| | | 128 | 8 | $\mathbf{89.61 \pm 1.63}$ | $\mathbf{13.19}$ |
| | 12.5% | 32 | 16 | $81.67 \pm 1.97$ | 16.08 |
| | | 64 | 8 | $\mathbf{82.87 \pm 1.82}$ | $\mathbf{16.00}$ |
| | | 128 | 4 | $82.10 \pm 2.06$ | 16.03 |

Table 1: Effect of SMYRF attention approximation on a pre-trained BigGAN (with no training). Rounds denote the number of LSH hashes and $C$ the number of queries per cluster.

(25% memory reduction, 5th row). We summarize results on Table 2. Remarkably, SMYRF-BERT (slightly) **outperforms** original dense attention, while using 50% less memory. We also underline that SMYRF-BERT with 25% of original memory, maintains $\approx \mathbf{99}$% of original model performance, while the BERT-model that uses the same memory (last row) maintains only $\approx 89$%.

To demonstrate that SMYRF scales for larger models, we also run experimetns with **SMYRF-BERT large** to a subset of the GLUE tasks. Specifically, SMYRF-BERT large obtains $\mathbf{60}.4$% performance (Matthew's Correlation) in the CoLA task and $\mathbf{90}.2$% (accuracy) in the QQP task. Both scores are significantly improved compared to the scores of the SMYRF-BERT base model, which shows that the approach scales to models with more attention layers. The corresponding scores of BERT large are $60.5$% and $89.3$% which are on par with the SMYRF performance for that model.

Since GLUE [25] datasets contain mostly short inputs, we also experiment on the IMDB [52] dataset, using sequence length $512$ tokens[3]. We experiment with SMYRF-BERT (base) and we report results for various configurations (memory savings, hashing rounds and cluster size). To support our argument that our algorithm is a drop-in replacement to *any* dense attention layer, we also include some results for RoBERTa [9] (base). Results are summarized in Table 3. Notably, SMYRF-BERT maintains $97.2$% of dense attention performance for $87.5$% memory savings.

In Table 4, we provide results for a Back-and-Forth procedure: we finetune with SMYRF and then for inference we use dense attention. By doing that, we observe performance almost equivalent to training with dense attention, while saving computational resources with SMYRF training. This indicates interchangeability between SMYRF and dense attention, which has not been previously reported. We use it to to train in a memory efficient manner and obtain maximum final performance.

| | Avg. | # | $C$ | CoLA | MNLI-m/mm | MRPC | QNLI | QQP | RTE | SST-2 | STS-B |
|---|---|---|---|---|---|---|---|---|---|---|---|
| BERT$_{128}$ | 82.69 | 1 | 1 | 57.83 | 84.43/**84.68** | **88.41** | **91.31** | 89.70 | 65.70 | **93.46** | 88.73 |
| SMYRF- | **83.12** | 2 | 32 | 58.79 | **85.02**/84.27 | 87.69 | 91.14 | **89.72** | **68.59** | 93.23 | **89.65** |
| BERT | 81.74 | 2 | 16 | **58.90** | 82.86/83.49 | 85.72 | 89.53 | 89.33 | 64.98 | 93.12 | 87.75 |
| BERT$_{64}$ | 81.57 | 1 | 64 | 58.80 | 82.34/82.47 | 87.02 | 90.48 | 89.69 | 61.73 | 93.00 | 88.64 |
| BERT$_{32}$ | 73.56 | 1 | 32 | 56.40 | 64.51/63.41 | 77.89 | 79.81 | 88.59 | 55.23 | 92.66 | 83.53 |

Table 2: Results on GLUE [25] (dev). # : hashing rounds. $C$ : the number of queries per cluster. SMYRF outperforms BERT while using $50\%$ less memory in each of the 12 attention layers.

| | Dataset | Memory | Accuracy | Rounds | Cluster |
|---|---|---|---|---|---|
| BERT | | 100% | **94.12**% | 1 | 512 |
| | | 50% | 92.64% | 8 | 32 |
| SMYRF- | | 25% | 92.52% | 16 | 8 |
| BERT | IMDB | 12.5% | 91.46 | 8 | 8 |
| | | 6.25% | 88.78% | 8 | 4 |
| | | 3.125% | 87.49% | 4 | 4 |
| RoBERTa | | 100% | **94.96**% | 1 | 512 |
| SMYRF-RoBERTa | | 50% | 93.72 | 8 | 32 |

Table 3: Finetuning BERT [6] (base) and RoBERTa [9] (base) on IMDB dataset for various configurations. For SMYRF models, we **train** and **evaluate** with SMYRF.

| | Dataset | Memory | SMYRF Inference | Accuracy |
|---|---|---|---|---|
| RoBERTa | | 100% | ✗ | **94.96**% |
| SMYRF- | | 50% | ✗ | 93.72% |
| RoBERTa | IMDB | | ✓ | **94.62**% |
| BERT | | 100% | ✗ | 94.12% |
| SMYRF- | | 50% | ✗ | 92.64% |
| BERT | | | ✓ | **93.54**% |

Table 4: Interchangeability of SMYRF and dense attention. We **train** with SMYRF and **evaluate** with *dense attention* for lightweight training and maximum performance.

## 4.3 Training from scratch

We also include experiments for networks trained from scratch. This shows that a non-pretrained model can learn with randomly initialized, SMYRF layers. Initially, the random weights produce less sparsity. However, the model quickly learns to create sparse attention maps and learning under our framework is possible. We use BigGAN [1] as the underlying model (see Appendix for details). We conduct our experiments on Celeba-HQ [29], which contains 30K images of celebrities at resolution $1024 \times 1024$. We choose Celeba-HQ because: (i) images are in resolution higher than $128 \times 128$, (ii) our budget is limited and Celeba-HQ requires much less training steps compared to ImageNet [37]. With SMYRF, we move attention from $64 \times 64$ resolution to $128 \times 128$ and train with $50\%$ less memory than dense attention. In Table 5, we report FID for BigGAN and SMYRF-BigGAN after 120K steps training on Celeba-HQ-128 (downsampled to $128 \times 128$). SMYRF-BigGAN *outperforms* BigGAN's FID by **3.95**%. Generated images from our model are shown in Figure 2. We finally move the attention layer to resolution $256 \times 256$ (65k length) and we successfully train on Celeba-HQ-256 for 120K steps on a single TPU v3-8. As far as we know, no other GAN has been trained with attention in higher resolution than this. Details and generated images are included in the Appendix.

|        | Resolution | Attention | Memory | Rounds | $C$ | FID |
|--------|------------|-----------|--------|--------|-----|-----|
| BigGAN | $128 \times 128$ | $64 \times 64$ | 100% | 1 | 4096 | 26.06 |
| SMYRF-BigGAN | | $128 \times 128$ | 50% | 4 | 2048 | **25.03** |

Table 5: Results on BigGAN training on Celeba-HQ-128 for 120K steps. Moving attention from $64 \times 64$ to $128 \times 128$ helps performance: FID decreases from 26.06 to **25.03**. Memory percentages in this Table have as reference the memory a dense attention layer would use at the given resolution.

| Model | IMDB (3 epochs) |
|-------|-----------------|
| SMYRF-RoBERTa | **93.7%** |
| E2LSH | 89.3% |
| Reformer | 88.7% |

Table 6: LSH ablation experiment. The E2LSH model corresponds to the SMYRF-RoBERTa model using the E2LSH [35] hashing scheme instead of the asymmetrical transformations. The Reformer model corresponds to running SMYRF-RoBERTa with the cross polytope LSH [53] scheme, which is used in the Reformer [18] paper.

## 4.4 Comparison with other efficient attention techniques

To validate the effectiveness of the proposed asymmetrical transformations, we replace SMYRF's hashing scheme with the E2LSH [35] scheme and the cross-polytope LSH [54] scheme of the Reformer and we evaluate all models on the IMDB [52] dataset, after training for three epochs. The results are summarized in Table 6. As shown, the asymmetrical transformations of SMYRF largely outperform all the other LSH schemes. This is expected since by design SMYRF tries to form clusters that maximize the inner products between queries and keys, while E2LSH and Reformer try to minimize euclidean distance and angular distance respectively, which is not the best objective when dealing with queries and keys with different vector representations and arbitrary norms.

To compare with the Longformer [28], we evaluate SMYRF on the Hyperpartisan News Detection [55] dataset. For this task, Longformer reports $94.8\%$ accuracy with 4096 context-length. SMYRF obtains **97.2%** performance while only using 512 tokens. Longformer slightly outperforms (for $\approx 1\%$) SMYRF in the IMDB dataset but it uses 8 times more tokens to achieve that. Unfortunately, the available RoBERTa [9] models have been trained with maximum positional embeddings at 512 tokens and thus we cannot determine whether bigger sequence lengths would favor SMYRF. Nevertheless, SMYRF performs on par with other efficient attention techniques without requiring any pre-training.

## 5 Related work

The fact that attention maps of pre-trained layers are sparse is well-known [15, 16, 3, 17, 56, 57]. Relevant research to our work includes efforts to leverage that sparsity by limiting attention of each element to a subset of the original sequence. [23] proposes to limit attention to a sliding window around each element. Even though this simple idea is a strong baseline due to locality, this method is usually outperformed [20, 18, 19] by data-driven methods for assigning to each query the keys it will attend to. One recent research work that performs well with pre-defined sparsity is Longformer [28]. Longformer has been shown to perform well in downstream tasks after pre-training for 65K gradient steps, resuming MLM training of a pre-trained RoBERTa [9] model. However, this work requires custom GPU kernels that do not transfer across hardware (i.e. are not efficient on TPUs). SMYRF differs from Longformer in other important aspects as well: (i) SMYRF does not require (even though it might help) further pre-training before finetuning on downstream tasks. Therefore, SMYRF is a drop-in replacement of dense attention, while Longformer [28] requires some adaptation of the original dense attention. (ii) More importantly, the fixed sparsification idea used in Longformer [28] is fundamentally different from our idea of using clustering to approximate attention and (iii) SMYRF can be used interchangeably with dense attention while Longformer cannot. As we showed, a trained SMYRF attention lower can be converted back to a normal dense attention layer during inference.

There are three research works that are very relevant to ours since they also propose data-driven attention within each group: (i) the Reformer [18], (ii) the Sparse Sinkhorn Attention [20] paper and

(iii) the Routing Transformer [19]. Reformer [18] changes the dense attention layer twofold: (i) it tights vector representations of queries and keys, (ii) it sets their norm to be equal to 1. Reformer is the first paper to propose LSH for clustering queries and keys. In Reformer, instead of using Asymmetric LSH, the authors use Angular distance LSH for clustering. This works because of (i), (ii), i.e. the Maximum Inner Product Search problem is equivalent to the Nearest Neighbor Search problem. We consider SMYRF as a generalized version of Reformer, since it employs Asymmetric LSH clustering to enable grouping of queries and keys that (i) do not have the same vectors, (ii) possibly live outside or inside the unitary $d-$dimensional disk. Apart from this, SMYRF and Reformer are similar: both networks sort vectors based on their LSH hash and both have linear attention complexity. Sinkhorn [20] proposes a differentiable sorting module for clustering queries and keys. The sorting layer is trained end-to-end with the rest of the model. It has only been shown to work well for training from scratch and not for fine-tuning of pre-trained models. Routing Transformer [19] proposes $k-$means clustering. In general, vectors that have small Euclidean distance are not guaranteed to have big inner product. To alleviate this, in Routing Transformer queries and keys are forced to have exactly the same vector representations and are also mapped to a $d-$dimensional unitary disk, exactly as Reformer proposed. Because of these changes, also this method cannot be applied to pre-trained models. Routing transformer has some other weaknesses as well: (i) the complexity is $O(N^{1.5})$ instead of $O(N \log N)$ which is the attention complexity of SMYRF and Reformer and (ii) the clusters are not guaranteed to be balanced. To solve (ii), [19] proposes to keep the top-k vectors in each cluster. However, this is not guaranteed to work well since it depends on the clusters ordering.

Comparing to the aforementioned methods, SMYRF is the only method that assigns dynamically queries and keys in clusters and can be applied to pre-trained models. Due to its portability, SMYRF is the first sparse attention model to report GLUE results on par with the underlying models. As we showed, SMYRF can be used interchangeably with dense attention before, during and after training. It also has linear attention complexity, similarly to Reformer. To the best of our knowledge, we are also the first to prove that the problem that all these methods are trying to solve is NP-hard.

The optimization problem that SMYRF tries to solve is connected to the problem of bi-clustering [58]. Indeed, as shown in the proof of Theorem 3, the goal in Attention Biclustering is to find a clustering of rows and columns of a matrix that maximizes the sum of the values of the clusters, where each value at position $(i, j)$ depends on the inner product of query $i$ and key $j$. For bi-clustering, iterative algorithms have been proposed [59]. Iterative techniques cannot be applied in the context of attention in which everything happens in a parallel fashion for fast execution in modern hardware.

Finally, there are a lot of others not attention related techniques that can be used to save memory and offer speedups. Examples of such techniques include knowledge distillation [60, 61], reversible layers [62], gradient checkpointing [63], quantization [64] and pruning [65, 66]. SMYRF and all these innovations are not mutually exclusive, i.e. they can be used together for maximum efficiency.

# 6 Conclusions

In this work we presented SMYRF, a novel type of balanced clustering to approximate attention. It is based on Asymmetric LSH with novel transformations and an adaptive clustering scheme. As it does not require changes to attention, SMYRF is the first sparse attention method that can be applied directly to pre-trained models. We showed powerful experimental results, in terms of performance, memory and speed. We also defined the underlying optimization problem that SMYRF tries to solve and we proved it is NP-hard. The strong experimental performance of SMYRF inclines us to believe that good approximation algorithms exist for this problem. Proving approximation guarantees for our method and discovery of better approximation algorithms are left for future work.

## 7 Broader Impact

Our main contribution is to reduce the computational requirements for machine learning models with attention-layers. Thus, any broader impact is likely to come from making these models more efficient in both memory impact and inference speed. We expect that this will be mostly a good thing since it democratizes the use of big attention layers: those who want to use such models but for whom the computational resources required are too great (like university labs) will now have an easier time. Moreover, GANs and language models will become easier to deploy on phones or other embedded devices. Further, more efficient training reduces the environmental and energy footprint of deep learning research. As the number of parameters of Transformer models grows, the latter becomes critical [67].

Negative consequences are also possible: The idea of DeepFakes [68] has been well-discussed elsewhere; a technique that makes these easier to create clearly has downsides. On the other hand, any sufficiently determined actor (e.g. a nation-state attempting to commit election-fraud) already has access to such technology, so perhaps the marginal negative impact will not be that large. Still, whenever computational requirements are reduced, the ease of taking bad actions increases along with the ease of taking good actions.

Finally, the technique proposed in this paper relies heavily on the assumption that attention maps are approximately sparse. It's possible (though we have no particular reason to think that this has happened or would happen) that, at some intermediate layer of a complicated neural network, enforcing sparsity when the ground-truth maps are non-sparse could result in ignoring salient features of atypical data points, thus resulting in fairness-related issues. Determining whether these approximations cause fairness issues in general could be an interesting subject for future work.

## 8 Acknowledgements

We would like to wholeheartedly thank the TensorFlow Research Cloud (TFRC) program that gave us access to v3-8 Cloud TPUs and GCP credits that we used to run our Computer Vision experiments. This research has been supported by NSF Grants CCF 1763702,1934932, AF 1901292, 2008710, 2019844 research gifts by Western Digital, WNCG IAP, computing resources from TACC and the Archie Straiton Fellowship.

## Footnotes

[1]The reported numbers are calculated by inspecting the attention maps of 1000 random generated images.

[2]Since BigGAN's official checkpoints are not publicly available, we use the authors' open-source, PyTorch [36] pre-trained models: https://github.com/ajbrock/BigGAN-PyTorch

[3]Note that for fair comparison with dense attention, we train SMYRF layers and dense layers on the same sequence length, following the comparison scheme of Reformer [18]. As previous work has shown [28], training on IMDB (and other long-input datasets) with bigger sequence length can help performance.

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
