[Supplementary Material]

# Supplementary Material
# SMYRF:
# Efficient Attention using Asymmetric Clustering

**Giannis Daras**
Computer Science Department
The University of Texas at Austin
`giannisdaras@utexas.edu`

**Augustus Odena**
Google Research
`augustusodena@google.com`

**Nikita Kitaev**
Google Research
`kitaev@cs.berkeley.edu`

**Alexandros G. Dimakis**
ECE Department
The University of Texas at Austin
`dimakis@austin.utexas.edu`

## 9   NP-hardness of Attention Biclustering

To prove Theorem 1, we first prove the following lemma.

**Lemma 1.** *The optimization problem:*

$$\min_{\mathcal{C}_t^L \in \mathcal{C}^L} ||\hat{P}_\epsilon - P||_F^2$$

*is NP-hard.*

*Proof of Lemma 1.* We will show that this problem is NP-hard, by showing that if we could solve in polynomial time all instances of this problem, we could solve in polynomial time the 3-dimensional matching problem (3-DM), which is known to be NP-complete. Following the notation of the main paper, we define $\epsilon = e^{-a}$ and $\hat{P}_\epsilon$ denotes the queries-keys product matrix with $-a$ in positions that correspond to queries and keys that do not belong to the same cluster.

It holds that:

$$\min_{\mathcal{C}^L} ||\hat{P}_\epsilon - P||_F^2 = \min_{\mathcal{C}_t^L \in \mathcal{C}^L} \sum_{(q,k) \notin \mathcal{C}_t^L} (q \cdot k - (-a))^2$$

$$= \min_{\mathcal{C}_t^L \in \mathcal{C}^L} \left[ \sum_{(q,k) \in \mathcal{Q} \times \mathcal{K}} (q \cdot k + a)^2 - \sum_{(q,k) \in \mathcal{C}_t^L} (q \cdot k + a)^2 \right] = \min_{\mathcal{C}_t^L \in \mathcal{C}^L} \left[ - \sum_{(q,k) \in \mathcal{C}_t^L} (q \cdot k + a)^2 \right]$$

$$= \max_{\mathcal{C}_t^L \in \mathcal{C}^L} \sum_{(q,k) \in \mathcal{C}_t^L} (q \cdot k + a)^2 \qquad (6)$$

Since for all given sets $\mathcal{Q}, \mathcal{K}$ we can create (in polynomial time) sets $\mathcal{Q}', \mathcal{K}'$ such that: $(q \cdot k + a)^2 = q' \cdot k', \quad \forall (q,k) \in \mathcal{Q} \times \mathcal{K}, (q',k') \in \mathcal{Q}' \times \mathcal{K}'$, the problem is equally hard to solving:

$$\max_{\mathcal{C}_t^L \in \mathcal{C}^L} \sum_{(q',k') \in \mathcal{C}_t^L} q' \cdot k' \qquad (7)$$

We can refer to this latest optimization problem as the max-mass problem.

Now consider the case where: $|\mathcal{Q}| = |\mathcal{K}|$, $L = |\mathcal{Q}|/2$, i.e. for this problem instance we have the same amount of queries and keys and we want to group them optimally to clusters with the constraint that each cluster should contain exactly 2 queries and 2 keys.

Note that for 1 query and one key per cluster this becomes weighted bipartite matching (which is efficiently solvable). For 1 query and $m$ keys per cluster this is a generalized matching problem, which is also polynomially solvable [69].

If we are able to solve the latter with a polynomial algorithm, then we can show that we can solve the 3-DM problem with a polynomial algorithm.

Any instance of the 3-DM problem can be expressed with finite, disjoint sets $X, Y, Z$ and a set $T$ of triples $(x, y, z)$ : $x \in X, y \in Y, z \in Z$. Visually, we can depict any instance of a 3-DM as a graph with three disjoint vertex sets, with $T$ containing the edges of the graph. For example, the 3-DM instance $X = \{1_{\text{red}}, 2_{\text{red}}\}, Y = \{1_{\text{blue}}, 2_{\text{blue}}\}, Z = \{1_{\text{green}}, 2_{\text{green}}\}, T = \{(1_{\text{red}}, 1_{\text{blue}}, 1_{\text{green}}), (1_{\text{red}}, 2_{\text{blue}}, 2_{\text{green}}), (2_{\text{red}}, 1_{\text{blue}}, 1_{\text{green}})\}$ is shown in (1,1) of Figure 3. We are looking for a set $T' \subseteq T$ in which every vertex is covered exactly once. Finding this solution, in case it exists, it is known to be an NP-hard problem. For this example, there is a valid solution, which is shown in (1, 2) of Figure 3.

We can transform any instance in the following way: we create one query and one key vector for each vertex $x \in X$ with the property that their inner product is some large positive constant $r_1 \in \mathbb{R}^+$. We can visualize this using red edges, following the previous example where we denoted with red color the vertices of $X$. We also set the inner product of any key vector that corresponds to vertex of $X$ with all the other query vectors to be $0$. Visually, a "missing" edge means that the inner product of the corresponding vectors is $0$ (no-reward). We also create a key vector for each vertex $y \in Y$ with the property that if $(x, y, z) \in T$ for some $z$, then the key vector for $y$ and the query vector for $x$ have inner product $r_1$, else $0$. We can show the non-zero edges of this category visually with blue color, following the previous example. Note that blue and red edges are equivalent in terms of the inner product between the vertices they connect, since both have inner product $r_1$. Finally, we create a query vector for each $z \in Z$ with the property that if $(x, y, z) \in T$ for some $x$ then the key vector for $y$ and the query vector for $z$ have inner product $r_2$, else $0$ where $r_2 \in \mathbb{R}^+$ is a small positive constant. Again, we can show the non-zero edges of this category with green color, following the previous example. For the given example, the transformation is shown in (2, 1) of Figure 3.

We have hypothesized that we have a polynomial algorithm to solve the max-mass problem of (7). The key observation for our proof is that, by construction, the best cluster in terms of potential accumulated mass is a cluster with one red, one blue and one green edge, as the ones shown right of the dashed bar of Figure 4. Indeed, the only way to obtain a cluster of more mass is to group two blue vertices with two red vertices, as shown in (1, 1) of Figure 4. By doing that, you earn one more $r_1$ compared to the clustering shown in (1, 2) of Figure 4, but you lose $2 \cdot r_1$, which are the rewards that they red keys could give (as they are left with no connections). Thus, the two clusterings on the right side of Figure 4 are preferable compared to any other potential two clustering that can be obtained by choosing the left grouping.

Since we have proved that the best possible clustering is one with one red, one blue and one green edge, it is now left to prove that if there is a 3-DM, then it is possible to group all queries and keys into clusters with this optimality property. Indeed, if there is a 3-DM, we can cover each vertex exactly one time, by matching any vertex of $X$ with a vertex from $Y$ and a vertex from $Z$. With our transformation, this means that we can group each red node with itself and one blue and one green vertex, which is an optimal cluster as it contains one red, one blue and one green edge. Thus, solving polynomially our problem would mean that we could also solve in polynomial time the 3-DM, which is known to be NP-hard.

$\square$

*Proof of Theorem 1.* We will show that if we can solve in polynomial time the problem: $\min_{\mathcal{C}^L} ||\sigma(\hat{P}_0) - \sigma(P)||_F^2$, then we can also solve in polynomial time the problem $\min_{\mathcal{C}^L} ||\hat{P}_\epsilon - P||_F^2$ (for an appropriate $\epsilon$) which we have proven to be NP-hard.

Figure 3: (1, 1): Instance of $3-$DM. We denote with red color the $X$ vertex set, with blue the $Y$ vertex set and with green the $Z$ vertex set. (2, 1): Ours transformation for the reduction. Red and blue edges have reward $r_1$, while green edges have reward $r_2 << r_1$. Missing edges have reward $0$. We create one query and one key for each vertex of $X$. We also create one key (blue color) for each vertex of $Y$ and one query (green color) for each vertex of $Z$. Connections between red queries and blue keys, as well as, connections between blue keys and green queries follow the problem instance. (2, 2): Optimal queries, keys clustering in groups of 2 for the max-mass 7 problem. (1, 2): Transformation of (2, 2) solution back to the $3-$DM instance.

Figure 4: Illustration of potential clusterings. (1, 1): sub-optimal clustering. (1, 2): optimal clusterings. Even though the clustering at the left side obtains more mass compared to any of the clusterings in the right side, it loses entirely the rewards that red keys can give. Indeed, clustering on the left side has one $r_1$ reward more than any of the two clusterings on the right, but in further clusterings red keys $\{1, 2\}$ will not be matched with anything (by construction) and thus a total reward of $2r_1$ will be lost.

We are given sets $\mathcal{Q}, \mathcal{K}$ and a number $L$. For each $q_i \in \mathcal{Q}$, we create a key vector $k_{q_i}$ such as
$q_j \cdot k_{q_i} = \begin{cases} a, & \text{if } i = j \\ -\infty, & \text{o/w} \end{cases}$ , where $a$ is a positive constant the choice of which we will determine later in this proof.

We denote the augmented key set with $\mathcal{K}'$.

We will now solve, with our hypothetical polynomial algorithm, the following optimization problem for our new input set:

$$\min_{\mathcal{C}^L} ||\sigma(\hat{P}_0) - \sigma(P)||_F^2$$

It holds that:

$$\min_{\mathcal{C}^L} ||\sigma(\hat{P}_0) - \sigma(P)||_F^2 = \min_{\mathcal{C}_t^L \in \mathcal{C}^L} \sum_{(q,k) \notin \mathcal{C}_t^L} \left( \frac{e^{q \cdot k}}{q_D} \right)^2 + \sum_{(q,k) \in \mathcal{C}_t^L} \left( \frac{e^{q \cdot k}}{q_D} - \frac{e^{q \cdot k}}{q_{\mathcal{C}_t^L}} \right)^2$$

$$= \max_{\mathcal{C}_t^L \in \mathcal{C}^L} \sum_{(q,k) \in \mathcal{C}_t^L} e^{2q \cdot k} \cdot \left( \frac{2}{q_D \cdot q_{\mathcal{C}_t^L}} - \frac{1}{q_{\mathcal{C}_t^L}^2} \right),$$

where $q_D$ denotes the denominator of the dense softmax and $q_{\mathcal{C}_t^L}$ denotes the denominator of the cluster softmax, i.e. $q_D = \sum_{k \in \mathcal{K}} e^{q \cdot k}$ and $q_{\mathcal{C}_t^L} = \sum_{k \in \mathcal{C}_t^L} e^{q \cdot k}$ for a given cluster $\mathcal{C}_t^L \in \mathcal{C}^L$.

We will now show that for a proper choice of $a$, this problem is equivalent to:

$$\max_{\mathcal{C}_t^L \in \mathcal{C}} \sum_{(q,k) \in \mathcal{C}_t^L} e^{2q \cdot k}.$$

Let $R_q = \frac{2}{q_D q_{\mathcal{C}_t^L}} - \frac{1}{q_{\mathcal{C}_t^L}^2}$. As we increase the value of $a$, the inner product of its query with its' special key gets significantly bigger compared to other inner products and thus for large enough values of $a$, we know that each query will get clustered with its' special key. We can control how close $q_D, q_{\mathcal{C}_t^L}$ are by setting appropriately the $a$ value. Specifically, we choose $a$ such that $q_D(1 - \epsilon) < q_{\mathcal{C}_t^L}$, $\forall q \in \mathcal{Q}$, $\mathcal{C}_t^L \in \mathcal{C}^L$, where $\epsilon = \epsilon(a)$ a small positive constant the choice of which we will determine soon. By definition, $q_{\mathcal{C}_t^L}$ is always smaller than $q_D$, and thus we for that choice of $a$ we have $q_D(1 - \epsilon) < q_{\mathcal{C}_t^L} < q_D$. Then, $R_q > \frac{2}{q_D^2} - \frac{1}{q_D^2(1-\epsilon)^2} = \frac{1}{q_D^2}(2 - \frac{1}{(1-\epsilon)^2}) = \frac{1-\epsilon'}{q_D^2}$ where $1 + \epsilon' = \frac{1}{(1-\epsilon)^2}$. But also, $R_q = \frac{2q_{\mathcal{C}_t^L} - q_D}{q_{\mathcal{C}_t^L}^2 q_D} < \frac{2q_D - q_D}{q_{\mathcal{C}_t^L}^2 q_D} = \frac{1}{q_{\mathcal{C}_t^L}^2} < \frac{1}{(1-\epsilon)^2 q_D} = (1+\epsilon')\frac{1}{q_D^2}$. Then, we have that:

$$\frac{1 - \epsilon'}{q_D^2} < R_q < \frac{1 + \epsilon'}{q_D^2}. \tag{8}$$

Now consider the following optimization problems:

$$\begin{cases} P_0: & \max \sum_{(q,k) \in \mathcal{C}_t^L} e^{2qk} R_q \\ P_1: & \max \sum_{(q,k) \in \mathcal{C}_t^L} \frac{e^{2qk}}{q_D} \end{cases}.$$

Let $F(c), G(c)$ the objective functions of $P_0, P_1$ respectively.

Using (8), we get that:

$$(1 - \epsilon')G(c) \leq F(c) \leq (1 + \epsilon')F(c). \tag{9}$$

Our claim is that for a suitable choice of $\epsilon'$, i.e. for a suitable choice of $a$, it holds that $\text{argmax } P_0 = \text{argmax } P_1$[4]. We prove that by contradiction. Let $c_1$ be the optimal choice of $P_0$ and $c_2$ be the optimal choice of $P_1$. Then, we know that $F(c_1) > F(c_2)$ and $G(c_2) > G(c_1)$. Using (9), we get that:

$$(1 - \epsilon')G(c_1) - (1 + \epsilon')G(c_2) < F(c_1) - F(c_2) < (1 + \epsilon')G(c_1) - (1 - \epsilon')G(c_2). \tag{10}$$

We denote with $d$ the gap between the optimal value $F(c_1)$ and the non optimal solution $F(c_2)$, i.e. $d = F(c_1) - F(c_2)$. Then, from (10), we get that:

$$d < (1 + \epsilon')G(c_1) - (1 - \epsilon')G(c_2) - (1 - \epsilon')G(c_1) + (1 + \epsilon')G(c_2) = 2\epsilon'(G(c_1) + G(c_2)).$$

Let $\theta_1$ the maximum value of $G(c_1) + G(c_2)$ among all the clusterings $c_1, c_2 \in C^L$, i.e. among all the possible valid clusterings in $L$ groups. Then, $d < 2\epsilon'\theta_1$. However, since $F$ is a function that maps from discrete clusterings to real numbers, two non-optimal solutions of $F(c)$ differ for at least a minimum distance. In that case, the minimum distance should be at least $e^{p_{\min}}R_{\min}$, where $p_{\min}$ is the minimum product between any query and any key and $R_{\min}$ is the minimum value that $R$ can take for any clustering. Let $\theta_2 = e^{p_{\min}}R_{\min}$. Then, $d \geq \theta_2$. If we choose $\epsilon'$ such that: $2\epsilon'\theta_1 < \theta_2$ then we have a contradiction. This is always possible since we can set the value of $\epsilon'$ to arbitrarily small values as we grow $a$ arbitrarily big. Thus, we proved that the problems $P_0, P_1$ have the same argmax for a proper choice of $a$. Then, for that choice of $a$ the problem $\min_{C^L} ||\sigma(\hat{P}_0) - \sigma(P)||_F^2$ is equivalent to $P_1$ which is equivalent to the problem:

$$\max_{\mathcal{C}_t^L \in \mathcal{C}} \sum_{(q,k) \in \mathcal{C}_t^L} e^{2q \cdot k},$$

since $q_D$ does not affect the choice of optimal clusters.

In the latter problem, we can replace all queries $q$ and keys $k$ with new vectors $q', k'$ such that: $q' \cdot k' = e^{2q \cdot k}$. This is equally hard to solving:

$$\max_{\mathcal{C}_t^L \in \mathcal{C}} \sum_{(q,k) \in \mathcal{C}_t^L} q \cdot k$$

which we proved to be NP-hard. $\qquad\qquad\qquad\qquad\qquad\qquad\qquad\qquad\qquad\qquad$ $\square$

## 10  Code

To encourage further research in sparse attention models, we open-source all our code and we release a Python package, named `smyrf`. The repository for the code is the following: https://github.com/giannisdaras/smyrf . `smyrf` implements SMYRF attention for Pytorch [36]. We plan to release implementation for Tensorflow [70] soon as well. `smyrf` contains various examples on pre-training and finetuning state-of-the-art models for Computer Vision and Natural Language Processing tasks. Regarding examples, at the moment `smyrf` includes:

- a TPU-compatible implementation of SMYRF-BigGAN, based on the official Pytorch implementation (https://github.com/ajbrock/BigGAN-PyTorch) for GPUs.
- code for training SMYRF-BigGAN on Celeba-HQ on a single TPU device.
- interactive notebooks showing how to use a pre-trained BigGAN for image generation with SMYRF on Celeba-HQ and ImageNet.
- tools to visualize cluster memberships for pixels of SMYRF generated images.
- code for replacing dense attention with SMYRF layers for state-of-the-art pre-trained NLP models, compatible with HuggingFace's Transformers [71] library.
- interactive notebooks for fine-tuning pre-trained NLP models on GLUE [25] and IMDB [52].
- tools for profiling SMYRF's performance compared to dense attention.

We also share the weights of SMYRF-BigGAN trained on Celeba-HQ at resolutions $128 \times 128$ and at $256 \times 256$ with attention at $128 \times 128, 256 \times 256$ respectively. Although these models are outperformed by non-attention GANs (e.g. StyleGAN [72, 73]), we believe that releasing them will help researchers understand better attention at higher resolutions. Hopefully, SMYRF will motivate the usage of more attention layers on new GAN architectures.

## 11  Singular values decay for pre-trained models

As noted in the paper, row-wise softmax can change the rank of a matrix. For example, the matrix $\begin{bmatrix} 1 & 0 \\ 2 & 0 \end{bmatrix}$ has rank 1, while the matrix $\sigma\left(\begin{bmatrix} 1 & 0 \\ 2 & 0 \end{bmatrix}\right) = \begin{bmatrix} 0.7311 & 0.2689 \\ 0.8808 & 0.1192 \end{bmatrix}$ has rank 2. Back to the context of attention, we have defined the product matrix $P = Q \cdot K^T$, where $Q : \mathbb{R}^{|\mathcal{Q}| \times d}$ represents the queries matrix and $K : \mathbb{R}^{|\mathcal{K}| \times d}$ the keys matrix. By the definition of rank, if the embeddings dimension is smaller than the sequence length dimension, i.e. $d < \min(|\mathcal{Q}|, |\mathcal{K}|)$, then P is low rank. However, the attention matrix after softmax, i.e. $\sigma(P)$, could be a full rank matrix. In this section, we provide experimental evidence that attention maps produced by pre-trained models are actually near low-rank.

Figures 5, 6 depict the singular values of the attention maps (for a random input[5]) for a pre-trained BigGAN (attention map dimensions: $4096 \times 1024$) and a pre-trained BERT (shown attention map dimensions: $64 \times 64, 256 \times 256$). For the pre-trained BigGAN (Figure 5) the decay in singular values is exponential. Specifically, in Figure 5 most singular values are very close to 0, which means that the attention map is effectively low rank. Figure 6 shows decay of singular values for a pre-trained BERT for sequence lengths: (a) 64, (b) 128. We illustrate decay for 144 heads (12 heads for each one of the 12 layers). For the majority of heads, singular values decay exponentially. We also see that the heads that do not demonstrate exponential decay in the singular values maintain this property

for both inputs (e.g. see the red line in both plots). In our experiments, we find that these heads are harder to approximate with SMYRF.

Singular values of BigGAN's attention map (after softmax) for 6 randomly generated images.

Figure 5: Decay of singular values of the attention map (after softmax) of a pre-trained BigGAN. Decay of singular values is exponential, which means that the matrix after softmax is effectively low rank.

## 12 Cluster memberships for generated images of a pre-trained BigGAN

In this section, we visualize how SMYRF's adaptive clustering algorithm assigns queries in clusters for a pre-trained BigGAN. This inspection gives useful insights into how the algorithm actually works in practice.

Top row of Figure 7 shows a random maltese dog generated by a pre-trained BigGAN [1]. The second row, illustrates how a single SMYRF hashing round assigns queries and keys for this particular image in two clusters: the first cluster is denoted with gray and the second with white color. As shown, SMYRF assignments preserve locality while enabling the modeling of arbitrary complex dependencies between input pixels. Indeed, pixels in the same neighborhoods are mostly organized in the same cluster. This observation is even more pronounced for background pixels (see big gray blocks). However, we also see that distant pixels sometime belong to the cluster as well. By only looking at the assignments in clusters (second row), we can infer that the image is roughly separated in three parts: the top part (mostly gray pixels), the middle part (mostly white pixels) and the bottom part (mostly gray pixels). These parts correspond to the top background, the dogs' face and the bottom background respectively. Third row of Figure 7 illustrates (for the same image) assignments in 128 clusters. Each cluster contains 32 queries and is denoted with a distinct color. Again, we observe that clusters are often local. Indeed, usually consecutive pixels or nearly consecutive pixels are denoted with the same color. For such large number of clusters, it becomes very difficult to extract semantic information from the clustering map without looking at the original image. However, by careful looking at both the attention map and the generated image we can make interesting observations. For instance, we see that distant background pixels are clustered together with much greater frequency compared to other distant non-background pixels. In other words, SMYRF often clusters together background pixels even if they belong to distant grid positions in the generated image (see for example colors in top and last row of the grid).

Singular values for 12 heads of 12 BERT attention layers.
Seq. length: 64

(a)

Singular values for 12 heads of 12 BERT attention layers.
Seq. length: 256

(b)

Figure 6: Decay of singular values for a pre-trained BERT for sequence lengths: (a) 64, (b) 128. We show decay of singular values 144 heads (12 heads for each one of the 12 layers). For the majority of heads, singular values decay exponentially. We also see that the heads that do not demonstrate exponential decay in the singular values maintain this property for both inputs (e.g. see the red line in both plots). We find that these heads are harder to approximate with SMYRF.

(a) Generated maltese dog from a pre-trained BigGAN.

(b) Visualization of SMYRF cluster assignments for this image (single hash). Total number of clusters: 2.

(c) Visualization of SMYRF cluster assignments for this image (single hash). Total number of clusters: 128.

Figure 7: Visualization of clustering assignments for a generated image by a pre-trained BigGAN.

# 13 SMYRF Clustering

## 13.1 Asymmetric Locality Sensitive Hashing (ALSH)

SMYRF clusters depend on the hashing indices of asymmetrically transformed queries and keys. As mentioned in the paper, we are looking for functions $F : \mathbb{R}^d \to \mathbb{R}^{d'}, G : \mathbb{R}^d \to \mathbb{R}^{d'}$ such as: $||F(q) - G(k)||_2^2 = D(q \cdot k), \; \forall (q, k)$ where $D : \mathbb{R} \to \mathbb{R}$ a decreasing function that depends only on the inner product $q \cdot k$. Essentially, functions $F, G$ are applied to queries and keys to convert the problem of Maximum Inner Product Search (MIPS) to Nearest Neighbor Search (NNS). For the latter problem, a lot lot of effective Locality Sensitive Hashing (LSH) functions have been proposed (e.g. [35, 74, 53]). The novel idea of converting MIPS to NNS is called Asymmetric Locality Sensitive Hashing (ALSH) and was first introduced in [32]. Since then, a lot of different asymmetric transformations have been proposed [34, 35, 33]. In this section, we show why previously proposed transformations are not suitable for our problem and how our novel asymmetric transformations, defined in Equation 4, relate to previous work.

We list the asymmetric transformations that have been widely used to convert a MIPS to NNS:

$$\begin{cases} [32]: F(q_i) = \left[ q_i; \frac{1}{2}, ...; \frac{1}{2} \right], \; G(k_i) = \left[ Uk_i; ||Uk_i||_2^2; ...; ||Uk_i||_2^{2^m} \right] \\\\ [33]: F(q_i) = [q_i; 0], \; G(k_i) = \left[ k_i; \sqrt{M_K^2 - ||k_i||_2^2} \right] \\\\ [34]: F(q) = \frac{M_K}{||q||_2} \cdot [q; 0], \; G(k) = \left[ k; \sqrt{M_K^2 - ||k||_2^2} \right] \end{cases}$$

where $M_K = \max_k ||k||_2$ and U a positive constant such as: $||U \cdot k_i||_2^{2^{m+1}} \to 0, \; \forall k_i \in \mathcal{K}$. The corresponding Euclidean distances of the transformed vectors are given below:

$$\begin{cases} [32]: ||F(q_i) - G(k_i)||_2^2 = ||q_i||_2^2 + \frac{m}{4} - 2Uq_i \cdot k_i + ||U \cdot k_i||_2^{2^{m+1}} \\\\ [33]: ||F(q_i) - G(k_i)||_2^2 = ||q_i||_2^2 + M_K^2 - 2q_i \cdot k_i \\\\ [34]: ||F(q_i) - G(k_i)||_2^2 = 2 \cdot M_K^2 - 2\frac{M_K}{||q_i||} \cdot q_i \cdot k \end{cases}$$

In all these transformations the Euclidean distance of the transformed vectors, i.e. $||F(q_i) - G(k_i)||_2$ decreases linearly with the inner product $q_i \cdot k_i$. However, an extra term, $p(||q_i||)$, appears. Indeed, these transformations were proposed for the case of a single query (e.g. a user) and multiple keys (e.g. movies) and for such applications $||q_i||$ is considered constant. On the contrary, for our setting, the transformations of [32, 34, 33] cannot be applied since $||q||_2$ is no longer a constant. To illustrate this better, consider the case where $q_1, q_2 \in \mathcal{Q}$ with $q_1 \neq q_2$ and $k \in \mathcal{K}$ a key such as: $q_1 \cdot k = q_2 \cdot k$. Since we are looking for big inner products, we expect to have transformations $F, Q : ||F(q_1) - G(k)||_2 = ||F(q_2) - G(k)||_2$. For [32, 33], if $||q_1||_2 < ||q_2||_2$ then $||F(q_1) - G(k)||_2 < ||F(q_2) - G(k)||_2$ and for [34]: $||F(q_1) - G(k)||_2 > ||F(q_2) - G(k)||_2$. Thus, all [32, 34, 33] do not satisfy our desired property, i.e. $||F(q_1) - G(k)||_2 = ||F(q_2) - G(k)||_2$. To solve this problem, we propose (see main paper) the novel asymmetric functions:

$$F(q_i) = \left[ q_i; 0; \sqrt{M_Q^2 + M_K^2 - ||q_i||_2^2} \right], \qquad G(k_i) = \left[ k_i; \sqrt{M_Q^2 + M_K^2 - ||k_i||_2^2}; 0 \right] \quad (11)$$

where we use the constants $M_Q = \max_{q_i} ||q_i||_2, \quad M_K = \max_{k_i} ||k_i||_2$, or any other upper bound on the norms. With this transformation, all queries and keys are mapped to a $(d + 2)$-dimensional ball with radius $\sqrt{M_Q^2 + M_K^2}$ and the distance of the transformed vectors decreases linearly with the inner product of the original vectors:

$$||F(q_i) - G(k_i)||_2^2 = 2 \cdot \left( M_Q^2 + M_K^2 - q_i \cdot k_i \right). \quad (12)$$

Note that the Euclidean distance of the transformed vectors depends only on the inner product of the original vectors and not on individual norms $||q_i||_2$ as in previous work.

## 13.2 Adaptive Clustering

The next step, after the asymmetric transformations, is to map the transformed queries $F(q)$ and keys $G(k)$ to real numbers, so that if $||F(q) - G(k)||_2$ is small, then $|h(F(q)) - h(G(k))|$ is also small with high probability, where $h : \mathbb{R}^{d'} \to \mathbb{R}$ is the mapping function. After mapping, we sort independently queries and keys based on their hash and we split them into groups of equal size. There are numerous hashing functions [54, 35, 74, 75] $h : \mathbb{R}^{d'} \to \mathbb{R}$ that belong to the LSH family that we can leverage to achieve that. One of the most widely adopted hash functions for locality sensitive hashing is E2LSH [35]:

$$h_{\text{E2LSH}}(u) = \left\lfloor \frac{(u \cdot a) + b}{r} \right\rfloor \tag{13}$$

where $a = (a_1, ..., a'_d) \in \mathbb{R}^{d'}$ with $a_i \in \mathcal{N}(0, 1)$ and $b \in \mathcal{U}(0, r)$ and $r$ is a scalar parameter which controls LSH sensitivity. Since we re-group vectors by sorting on their LSH index, the floor operator and the division with $r$ are not needed. Our simplified hashing function is defined as:

$$h_{\text{ours}}(u) = (u \cdot a) + b \tag{14}$$

We roughly removed a division by a constant. Thus, this simplified hashing function preserves the locality-sensitive properties of E2LSH [35]. Namely, if $||u_1 - v_1||_2 \leq ||u_2 - v_2||_2$ then with high probability: $|h(u_1) - h(v_1)| \leq |h(u_2) - h(v_2)|$, $\forall u_1, u_2, v_1, v_2 \in \mathbb{R}^{d'}$.

## 13.3 Merging hashing rounds

In our experiments, we run multiple hashing rounds each time, similarly to [18]. Each time we run LSH, we end up with a (possibly) different clustering assignment and thus (possibly) different attention output. Specifically, we repeat the process $H$ times (where $H$ is usually a small constant, e.g. 8) to reduce the probability that we miss big inner products. In this section, we explain how we merge the partial attention outputs (made from different hashing rounds) into a single attention output.

Without loss of generality, we will present the merging algorithm for a single query $q$. At each clustering round $h$ we get (from the adaptive clustering) a set of key vectors $\mathcal{K}_{h_q} \subseteq \mathcal{K}$. The corresponding attention output is:

$$o_q^h = \sum_{k \in \mathcal{K}_{h_q}} w_k v_k, \qquad w_k = \frac{e^{q \cdot k}}{\sum_{k' \in \mathcal{K}_{h_q}} e^{q \cdot k'}}$$

We merge the attention outputs of the different rounds with a weighted sum. The weight, $a_h$, for each round $h$, is the fraction of the softmax mass that was acquired in this round to the total mass acquired by all rounds. Formally the attention output $o_q'$ for query $q$ is computed as:

$$o_q' = \sum_{h=1}^{H} a_h \cdot \sum_{k \in \mathcal{K}_{h_q}} w_k v_k, \qquad w_k = \frac{e^{q \cdot k}}{\sum_{k' \in \mathcal{K}_{h_q}} e^{q \cdot k'}}, \qquad a_h = \frac{\sum_{k' \in \mathcal{K}_{h_q}} e^{q \cdot k'}}{\sum_{n=1}^{H} \sum_{k' \in \mathcal{K}_{n_q}} e^{q \cdot k'}} \tag{15}$$

To explain this merging scheme, we will show that under certain assumptions, this merging scheme can lead to exact approximation of the real attention output. We start by listing these assumptions.

**Assumption 1** (Sparsity of weights). *For any given query $q \in \mathcal{Q}$, the key set $\mathcal{K}$ has at most $T$ and at least one vectors $k_i \in \mathcal{K}_q$ such as:*

$$k_i \in \mathcal{K}_q, \ k_j \notin \mathcal{K}_q \Rightarrow \frac{e^{q \cdot k_j}}{e^{q \cdot k_i}} = 0$$

From Assumption 1, it follows that at most $T$ and at least one key vector $k_i$ gets a non-zero score, $w_i \neq 0$, after softmax.

**Assumption 2** (Fairness of LSH clustering). *For any given query $q \in \mathcal{Q}$ and two keys $k_1, k_2 \in \mathcal{K}$, if $w_{k_1} \neq 0 \ \wedge w_{k_2} \neq 0$, then $\sum_{n=1}^{H} \sum_{k_1 \in \mathcal{K}_{n_q}} 1 = \sum_{n=1}^{H} \sum_{k_2 \in \mathcal{K}_{n_q}} 1$ where $H$ denotes the hashing rounds and $\mathcal{K}_{n_q}$ denotes the chosen key set for query $q$ at hash round $n$.*

Assumption 2 simply states that each query is clustered the same number of times with all its' big inner products along the different hashing rounds.

**Assumption 3** (Effectiveness of LSH clustering). *There is a small constant $H$, which denotes the number of hashing rounds, such as:*

$$\forall k \in \mathcal{K} : \; w_q \neq 0 \Rightarrow \exists n : \; 1 \leq n \leq H \; \wedge \; k \in \mathcal{K}_{q_n}.$$

The latter assumption states that we need a small number of hashing rounds $H$ to catch all big inner products of a given query.

We state the following theorem:

**Theorem 2.** *If Assumptions 1, 2, 3 hold, then our approximation algorithm is exact.*

*Proof of Theorem 2.* With our merging scheme (Equation 15), the attention output is:

$$o'_q = \sum_{h=1}^{H} \sum_{k \in \mathcal{K}_{h_q}} \left( \frac{\sum_{k' \in \mathcal{K}_{h_q}} e^{q \cdot k'}}{\sum_{n=1}^{H} \sum_{k' \in \mathcal{K}_{n_q}} e^{q \cdot k'}} \cdot \frac{e^{q \cdot k}}{\sum_{k' \in \mathcal{K}_{h_q}} e^{q \cdot k'}} \right) \cdot v_k = \sum_{h=1}^{H} \frac{\sum_{k \in \mathcal{K}_{h_q}} e^{q \cdot k} \cdot v_k}{\sum_{n=1}^{H} \sum_{k' \in \mathcal{K}_{n_q}} e^{q \cdot k'}} \tag{16}$$

Under Assumption 1, the dense attention output for this query is the vector:

$$o_q = \sum_{k \in \mathcal{K}_q} \frac{e^{q \cdot k}}{\sum_{k' \in \mathcal{K}_q} e^{q \cdot k'}} \cdot v_k$$

where $K_q$ is the set of keys $k_i$ for query $q$ for which $w_i \neq 0$.

Under Assumption 3, all keys that have big inner product with a given query $q$ are clustered with that query, at least one time. Also, under Assumption 2, all these keys are clustered the same amount of times with each query. We will denote the amount of a query is clustered with each one of its' big inner products with $N_q$. It holds that:

$$\sum_{n=1}^{H} \sum_{k' \in \mathcal{K}_{n_q}} e^{q \cdot k'} = N_q \cdot \sum_{k' \in \mathcal{K}_q} e^{q \cdot k'} \tag{17}$$

By substitution in Equation 17, we get:

$$o_q = \frac{\sum_{n=1}^{H} \sum_{k \in \mathcal{K}_{h_q}} e^{q \cdot k} \cdot v_q}{N_q \cdot \sum_{k' \in \mathcal{K}_q} e^{q \cdot k'}} \tag{18}$$

Under Assumptions 1, 2 small inner products get a zero-score and all big inner products are clustered $N_q$ times each. Thus, we can write for the nominator: $\sum_{n=1}^{H} \sum_{k \in \mathcal{K}_{h_q}} e^{q \cdot k} \cdot v_q = N_q \sum_{k \in \mathcal{K}_q} e^{q \cdot k'}$.

Substituting to Equation 18, we get:

$$o'_q = \sum_{k \in K_q} \frac{e^{q \cdot k}}{\sum_{k' \in \mathcal{K}_q} e^{q \cdot k'}} v_k = o_q$$

$\square$

In this section, we explained in detail our merging scheme. We also showed that under certain assumptions on the data, this scheme leads to exact approximations of dense attention output. We fully understand that the assumptions are far too tight to hold in practice and since distortion is introduced. However, as we demonstrated in the Experiments section, the distortion is negligible even for large memory reductions, since SMYRF can perform on par (or even better, e.g. GLUE) with dense attention, especially on downstream Natural Language Processing tasks, using a fraction of the original memory.

# 14 Complexity analysis and speedups

In the paper, we presented shortly the complexity of our algorithm. In this section, we explain it in more detail and we also include speed plots that demonstrate the effectiveness of SMYRF for long sequences.

## 14.1 Complexity Analysis

For the complexity analysis, we assume for simplicity that $|\mathcal{Q}| = |\mathcal{K}| = N$, i.e. the number of available queries is equal to the number of available keys.

We run the algorithm $H$ times (i.e. rounds of LSH). Each run has two stages:

- Clustering in L clusters (of equal size). For clustering, we hash all points with LSH which requires complexity $O(N)$ and then we sort points based on their hash, which requires complexity $O(N \cdot \log N)$. Overall, the complexity is $O(N \cdot \log N)$.
- Within clusters attention. Attention within each cluster has quadratic cost with respect to the cluster size. Each cluster has size $\frac{N}{L}$, so the complexity of attention in a single cluster is $O(\frac{N^2}{L^2})$. We have $L$ such clusters, and thus the overall complexity is $O(\frac{N^2}{L})$.

The total complexity is: $O\left(H \cdot N \cdot \log N + H \cdot \frac{N^2}{L}\right)$. We choose $L = O(N)$, i.e. each query attends to a small constant number of keys. We obtain complexity: $O(H \cdot N \cdot \log N)$.

## 14.2 Speedups

In this subsection, we present two speed plots to demonstrate the speed effectiveness of SMYRF for large sequences. The first plot, Figure 8, shows elapsed time for SMYRF-BERT (base) GPU inference for various batch-sequence length configurations. In all these experiments batch size $\times N = 65\text{K}$, where $N$ denotes the sequence length. We underline that SMYRF has (almost) constant speed in all these configurations while the speed of dense attention decreases rapidly us the sequence length increases. Notably, SMYRF is already faster than dense attention in sequence length 1024 tokens. The second plot, Figure 9, shows seconds per iteration for SMYRF-BERT (base) GPU inference for various hashes-cluster configurations. In all these experiments, batch size is fixed to 1. As shown, all different configurations significantly outperform (in terms of speed) dense attention as the sequence length increases.

# 15 Experimental details

## 15.1 Natural Language Processing experiments

In this section, we provide some details about the experimental settings for the Natural Language Processing experiments.

### 15.1.1 IMDB

IMDB [52] contains 25,000 train and 25,000 dev labeled movie reviews. The task is to identify if a given movie review is positive or negative. The average sentence length in IMDB is 300 tokens and the 95th percentile of context length is 705 tokens. In our experiments, we truncated/padded all sentences to 512 tokens. For all our experiments, we trained for 3 epochs, with batch size 8. We used Adam [76] as our optimizer with learning rate $3 \cdot 10^{-5}$. The dataset is available publicly in this link: https://ai.stanford.edu/ amaas/data/sentiment/. The experiments on IMDB run on a single GPU provided by Google Colab.

### 15.1.2 GLUE

GLUE [25] is a standard multitask benchmark for Natural Language Processing. For a full description of tasks, dataset statistics and files, please refer to the official website: https://gluebenchmark.com/. Following previous literature (e.g. [9, 6, 5, 24]), for our GLUE experiments we truncate/pad all

Figure 8: Elapsed time for SMYRF-BERT (base) GPU inference for various batch-sequence length configurations. Elapsed time for SMYRF is almost constant for all configurations. Elapsed time for dense attention worsens a lot as we increase the sequence length.

input sentences to 128 tokens. For GLUE, we trained for 3 epochs at batch size 16, warming up for 10% of the total training time. The learning rate was selected via grid search among the values $\{5 \cdot 10^{-5}, 3 \cdot 10^{-5}, 2 \cdot 10^{-5}\}$. We run the GLUE experiments on TPUs.

## 15.2 Training SMYRF-BigGAN on Celeba-HQ

In the paper we presented results for training SMYRF-BigGAN from scratch on Celeba-HQ [29]. As explained, we trained on Celeba-HQ (and not ImageNet [37]) in order to save computational resources. In this section, we provide the details for these experiments. First of all, as the name suggests, we used as the underlying model, BigGAN [1]. For our experiments we disabled BigGAN's hierarchical latent codes, shared embeddings and skip-z connections since Celeba-HQ has one single class (humans) and these architectural choices were introduced to model multiple classes (e.g. 1000 classes on ImageNet). We also found that for the single-class Celeba-HQ we didn't have to use very large batch sizes for stable training. For all our experiments, we used batch size 32. Following the BigGAN paper, we used Two Time Scale Update Rule (TTUR) [39] with Adam [76] optimizer, $G_{lr} = 2 \cdot 10^{-4}$, $D_{lr} = 5 \cdot 10^{-5}$, $\beta_1 = 0$ and $\beta_2 = 0.999$.

BigGAN for resolutions $\{128 \times 128, 256 \times 256\}$, is trained with a single attention layer at resolution $64 \times 64$. The authors mention that they stick attention to low resolution to save computational resources. We take advantage of SMYRF's reduced memory requirements to train with attention at resolution $128 \times 128$ and $256 \times 256$. For both experiments, we remove the dense attention layer and we add a SMYRF attention layer. Since our goal is to demonstrate the ability of SMYRF layers to train successfully from scratch, there is no reason to use higher (image) resolutions than the attention resolution and thus SMYRF is the final layer (before Tanh [77]) in the architecture. In other words, we train on image resolutions $128 \times 128, 256 \times 256$ respectively. Training on resolution $128 \times 128$ has the side-benefit that we can compare directly with the original BigGAN model (with dense attention at $64 \times 64$). As we demonstrated in the Experiments section of the paper, moving attention

Elapsed time per iteration on BERT (base) for different SMYRF configurations.
Batch size=1 for all experiments.

Figure 9: Seconds/iteration for SMYRF-BERT (base) GPU inference for various hashes-cluster configurations. In all experiments batch size is fixed to 1. SMYRF has approximate the same speed with dense attention at 1024 tokens. However, as the number of tokens increases, SMYRF is significantly faster than dense attention.

from $64 \times 64$ to $128 \times 128$ can lead to $\approx 4\%$ FID [39] improvement after 120K training steps[6]. We present random generated images from SMYRF-BigGAN with attention at resolution $256 \times 256$ at Figure 10. As explained in the Things that did not work section, training with SMYRF from scratch is harder as the sequence length increases. The main reason is that during the early stages of training attention maps are not sparse and thus our approximation's algorithm output is not close to the dense attention output. We noticed that the overall performance of SMYRF-BigGAN-256 is lower compared to SMYRF-BigGAN-128 and the generated images seem slightly less realistic. Despite the aforementioned shortcomings, this experiment demonstrated that it is possible to successfully train an attention GAN with attention at $256 \times 256$ resolution on a *single* TPUv3-8 device. The training at $128 \times 128$ resolution lasts approximately 1.5 day and at $256 \times 256$ resolution approximately 2 days.

## 16   Things that did not work

In this section, we discuss some negative results we encountered in the process of writing this paper. Our goal is to share our experience with the research community about the observed shortcomings of some approaches so that future research can re-formulate them, reject them or even contradict our findings. We also include some suggestions on potential ways to alleviate such problems that we did not have the time to explore in this paper.

Figure 10: Generated images from SMYRF-BigGAN on Celeba-HQ-256. Attention at $256 \times 256$. The trained model uses $50\%$ less memory compared to the memory dense attention would use.

## 16.1 Learning from scratch under extreme sparsity

Our initial goal was to train a SMYRF model from scratch with extreme memory reductions, e.g. to the magnitude of $99\%$. Such reduction could enable the training of SMYRF-BigGAN with attention at $1024 \times 1024$. However, our preliminary experiments with BigGAN [1], failed (mode-collapse very early in the training process). We tried to investigate this further and we found that during the early stages of the training the Frobenius norm of the difference between the SMYRF and the dense attention map is really high. We believe that this is due to the non-sparsity of the attention maps in the early stages of the training. It is also possible that their eigenvalues decay slower which means that their effective rank is higher compared to pre-trained models. One way to solve the problem is to dynamically adapt the memory reduction (e.g. by selecting the number of hashes) during the training. One way to achieve that is to use as many hashes as need to achieve a certain bound for the Frobenius norm. In the early stages of training, we expect that more hashes are needed for an accurate reconstruction. The number of hashes should decay as the training progresses and the attention maps become more sparse and have lower rank. One disadvantage of this approach is that at the early stages of the training, more memory is needed. However, we observed that the period of time in which the attention maps are not very sparse is minor compared to the whole training time for BigGAN and thus this approach can lead to significant savings. We aim to explore this more in the future.

## 16.2 Better LSH based clustering schemes

The biggest advantage of clustering with an LSH-based scheme is that the attention complexity is linear (compared to K-means clustering for example, see Routing Transformer [19]). However, while inspecting SMYRF, we found that LSH-clustering is the biggest bottleneck to greater performances. For example, if each query attends to at its' top-k (in terms of inner product) keys (instead of the keys assigned with LSH), the performance improves considerably. Finding exactly the top-k keys for each query is expensive (especially in high dimensions) and thus this approach is not viable. However, this observation motivates research in finding even more effective LSH-based clustering schemes. Even though we tried other ALSH variants, we did not manage to find something that works better than our proposed transformations till now. We consider this problem an interesting future direction since ALSH has been widely explored only for the case of a single query and multiple keys. In this paper, we did the first step in extending this to multiple queries, but we are inclined to believe that further research can lead to even better results in this direction.

## Footnotes

[4] We assume that if there is a set of optimal solutions, then we pick with the same order from that set for both problems.

[5]We experimented with different random inputs and there is no qualitative difference in the decay of singular values)

[6]We note that in order to save computational resources we stopped training for both models on 120K iterations, before mode-collapse. That means that further training could possibly lead to even better FID scores for both models.