[Reviews · NeurIPS 2020]

Review 1

Summary and Contributions: The paper describes a method for reducing the size of attention maps using LSH based hashing. As motivation for the approach, the authors describe that attention maps of trained models are usually rather sparse and that mapping keys and values to cluster, innerproducts between the corresponding clusters can replace the products of keys and values. The paper also shows that finding an optimal assignment is NP-hard and thus, heuristic approaches must be persued. To cluster the data key and values are mapped into a metric space where finding maximal inner products corresponds to kNN search. After applying an assysmtric local sensitive hash function keys and values can be partitioned into L equal sized clusters. The method can be applied to replace pretrained dense attention layers and can be used for training from scratch. However, this only seems to work well given that the dense attention map has a certain level of sparsity and the number of clusters is sufficiently large enough. These results show performance can be kept up even under considerable memory decrease.

Strengths: The paper addresses a major performance problem of attentions layers which is the quadratic size of attention maps. Results indicate that proposed method is capable to reduce memory requirements and keep up performance on two state-of-the-art architectures BigGAN and BERT. A final advantage of the method that it can be used for from scratch training. However, as mentioned in the appendix this is often problematic due to the dense nature of attention maps in early state of training.

Weaknesses: The evaluation of the method does not compare to any other method for reducing the memory requirements of attention layers like the closely related Reformer. The authors provided the comparison in the author feedback and they should add this analysis to the final version if accepted.

Correctness: I did not find any flaws in the underlying theoretic resutls.

Clarity: The paper is well understandable. However, for following the details reading the appendix helped me a lot.

Relation to Prior Work: The paper sufficiently highlights it differences to previous works. Comparison to the related methods were provided in the author feedback and should be included in the paper.

Reproducibility: Yes

Additional Feedback: ==================== post feedback ==================== I followed the author feedback as mentioned before I would like to see the providedd comparision to previous work in the paper.


Review 2

Summary and Contributions: The authors propose a novel type of balanced clustering algorithm to approximate attention. It uses Locality Sensitive Hashing (LSH) in a novel way by defining new Asymmetric transformations and an adaptive scheme that produces balanced clusters. The method can be directly used for pre-trained models and achieves competitive/better performance with BigGAN/BERT/RoBERTa by shrinking 50% memory.

Strengths: 1. The proposed model can save memory for both training and inference. The experiments are significant compared to the memory usage of BigGAN and BERT. 2. The proposed model can be widely applied for both CV and NLP tasks. 3. The analysis of the number of queries per cluster in the proposed model is quite interesting.

Weaknesses: 1. There is no ablation study for Novel Asymmetric Transformations. It unclear whether the proposed method is better than only using Locality Sensitive Hashing (LSH). 2. The proposed Adaptive Clustering still relies on LSH. Although there is some further optimisation on E2LSH to create balanced clusters, the method is still quite similar to Re-Former work. The authors need to have some fair comparison with Re-Former or LSH. 3. The proposed method can not significantly boost the performance. I would like to see whether the method can be applied on longer sequence which can not be encoded by BERT, such as LongFormer work. For the models of BERT/RoBERTa on GLUE, it seems no need to save memory for training. 4. I would like to see some analysis of reversible Transformer/CNN (reformer) and distillation works which can also save memory for either training or inference.

Correctness: Yes

Clarity: Yes

Relation to Prior Work: Yes

Reproducibility: Yes

Additional Feedback:


Review 3

Summary and Contributions: [EDIT: thank you for the response, for clarifying the proof, and for the additional experiments. I am increasing my score.] This paper presents an asymmetric LHS clustering strategy for speeding up neural attention to O(N log N). The key property is the "drop-in replacement" nature of the proposed layer, decently preserving performance when replacing a dense attention module with no fine tuning.

Strengths: - Drop-in nature makes proposed method much more useful than alternatives. - Strong significance: Efficient attention is highly relevant and sought after at the moment. - Evaluation with great results on image&text data in multiple regimes: with/without fine-tuning, training from scratch, several compression levels.

Weaknesses: - Experiments do not compare against related efficient attention models (reformer, routing transformer, longformer) even in settings where the comparison would be possible. - Phrasing and proof of Theorem 1 is a bit confusing and seems of little impact to the overall goals of the paper.

Correctness: To my assessment the proposed algorithm is correct. The "catch" seems to be knowing the max query and key norms up front, which is not available in the standard (A)LSH streaming setup, but is perfectly fine for attention. Some small doubts about Theorem 1 below, but IMO this result is not required for this paper. - Calling Theorem 1 and the associated problem "approximate attention clustering" is a bit confusing because it sounds like _approximating_ the optimization problem is NP-hard, rather than solving it exactly. Why not just "attention clustering" or "attention co-clustering", if connections to co-clustering turn out to be relevant? - In Lemma 1, it's not clear to me that the proofs works for eps=0, since you would need to subtract infinities, and in fact this seems to be the only reason you need to introduce eps at all, rather than to just mask everything to -inf. However, you need to set eps=0 to get to Theorem 1. Please check. - Right before line 88 (supplementary material) I believe you forgot a minus sign and the last line (as well as the eqns on line 90 and 91) should be max, not min?

Clarity: The paper is mostly clear and easy to read, with some minor concerns brought below. (4/5) - Figure 1 does not bring much clarity IMO and the letters in the circles are too small to read. Perhaps it's clearer to show a clustered attention weight matrix with some markings (or permuted row-cols)? - Expression (2) is a bit hard to read. It may be easier to define cluster assignments as a function from Q u K to cluster indices, rather than this reverse formulation. (See also notations in literature on biclustering.) - You need L=O(N) to get NlogN complexity, but the paper is not clear about how L is chosen. From the code & supp it seems to me you parametrize the *cluster size*, please clarify this in the main paper.

Relation to Prior Work: - Relationship to efficient attention is very clearly discussed. - Performance is not benchmarked against the previous contributions. - Missing connections to the literature on biclustering / co-clustering, which seem to deal with similar optimization problems -- please check and highlight this connection. For instance: Biclustering of Expression Data, Y. Cheng & G.M.Church, ISMB-AAAI, 2000.

Reproducibility: Yes

Additional Feedback: This paper seems to be a valuable contribution to the "efficient attention" body of work, specifically appealing for the drop-in potential. It is hard to say whether any performance is lost due to this drop-in potential: comparisons against e.g. Reformer would address this. On line 40 you point out that Star Transformer prevents causal masking. Does your architecture support causal masking? I appreciate your discussion of broader impact, especially the final concern that block-wise attention may introduce some hard-to-characterize bias. This seems worth thinking more about. Presentation and typos: - Table 2 is too wide - L245 "is drop-in" -> "is a drop-in" - L253 tight -> ties (? not sure what you meant) - L273 its' -> its - Check capitalization in references (e.g. lsh (alsh) is lowercase) - What is q_D, Q_C_D and q_C in the equation between lines 87-88 of the appendix? - The plots in the supplementary material are very low resolution, please use vector graphics (e.g. pdf). Plots like Fig. 3/4 in Supplementary should be log-scaled probably.


Review 4

Summary and Contributions: The paper proposes a balanced clustering method to approximate attention based on Asymmetric LSH and an adaptive clustering scheme. The work also proves that the underlying optimization problem is NP-hard.

Strengths: - The model does not require changes to attention. - The memory and speed are improved compared with the original self-attention module.

Weaknesses: - The results of the BERT experiments are not strong enough. Most gains come from the RTE that is a very small dataset. However, on the most important task MNLI, the performance degrades significantly. In terms of performance, it seems that knowledge distillation outperforms the proposed method, with the same reduction of memory size. - The speedup in terms of detailed GPU hours shouls also be reported in the tables. - The previous attention clustering methods are not throughly compared with in the experiments. - It's unclear how GPU friendly the method is. - More analysis would help readers to understand the limitation of the proposed method. - The method is evaluated for fine-tuned BERT models. It's more insightful to show that the proposed method can also work well in the pre-training setting.

Correctness: - The experiments should also be evaulated on the large-size BERT, where the attention approximation should be harder. It's unclear whether the proposed method performs worse along with the increased model size. - The results of the BERT experiments are not strong enough. Most gains come from the RTE that is a very small dataset. However, on the most important task MNLI, the performance degrades significantly. In terms of performance, it seems that knowledge distillation outperforms the proposed method, with the same reduction of memory size.

Clarity: The presentation is easy to follow in general. The font size in Figure 1 is too tiny.

Relation to Prior Work: - The previous attention clustering methods are not throughly compared with in the experiments. [1] Efficient Content-Based Sparse Attention with Routing Transformers [2] Reformer: The efficient transformer.

Reproducibility: Yes

Additional Feedback: ====after author response I've read the author response and increased my score to 6.

[Author Response · NeurIPS 2020]

| Experiment | Dataset | Model | Tokens | # | C | Perf. |
|---|---|---|---|---|---|---|
| Comp. with Longformer | Hyperpartisan news | Longformer | 4096 | NA | NA | 94.8% |
| | | SMYRF-RoBERTa | 512 | 4 | 64 | **97.2%** |
| Larger models | CoLA \| QQP | SMYRF-BERT (**large**) | 128 | 2 | 32 | 60.4% \| 90.2% |
| Comp. with Reformer / LSH Ablation | IMDB (3 epochs) | SMYRF-RoBERTa | 512 | 8 | 32 | **93.7 %** |
| | | E2LSH | 512 | 8 | 32 | 89.3% |
| | | Reformer (Cross-Polytope LSH) | 512 | 8 | 32 | 88.7% |

**New Experiments:** We thank the reviewers for their valuable comments! We see that reviewers appreciated the
advantages, especially that we can use this method as a drop-in replacement in pre-trained NLP and CV models.

[R1], [R2], [R3], [R4]: in response to your requests for further comparisons, we first compare with the Longformer,
which reports scores on downstream tasks. As shown in the table, we significantly outperform the Longformer on
Hyperpartisan News, even with significantly smaller context length. [R2]: *"There is no ablation study for Novel*
*Asymmetric Transformations. Unclear if the proposed method is better than only using LSH. [..] The authors need to*
*have some fair comparison with Reformer or LSH"*. Thank you for the suggestions. As shown in the table, our novel
ALSH significantly outperforms the E2LSH and the Reformer LSH scheme. [R3]: *"can the method be applied on*
*longer sequences?"* Yes. In fact, we managed to train BigGAN from scratch with **16K tokens** (see Table 5) and **65k**
tokens (see section 4.3 , 7.2 of supp.). Unfortunately, many pre-trained NLP models have been trained with maximum
positional embeddings at 512 tokens, which prohibits finetuning in larger inputs. This is one of the main reasons we
could not directly compare with Reformer / Routing Transformer. [R4]: *"It's unclear whether the proposed method*
*performs worse along with the increased model size."* We run additional experiments on some GLUE tasks to show that
our method works well even for bigger model sizes (see above Table). SMYRF-BERT large consistently outperforms
SMYRF-BERT base (see also Table 2). We used SMYRF on all 24 attention layers vs 12 for base. [R4]: *"Most gains*
*come from RTE which is a small dataset [..], on the most important task MNLI, the performance degrades significantly"*.
The dependency between dataset size and performance is unclear. For example, QQP is a fairly big dataset, in which
SMYRF outperforms vanilla BERT while using 50% less memory (see Table 2). We respectfully disagree that our
method creates a significant performance degradation on MNLI: with 50% less memory our reported score is less than
1% shy of the performance of BERT. To further address this, we performed hyperparameter search for MNLI (lr=$3e-5$,
batch=8, 5 epochs) and we obtained **85.02%** acc., which is *better* than the $84.43\%$ acc. of BERT. We plan to include
and expand the results of the above table in the Camera Ready version.

**On Distillation:** [R2], [R4] mentioned distillation as an alternative to reduce memory. Knowledge distillation creates
smaller models while our method allows larger inputs. The two innovations are not mutually exclusive. If we still
compare, DistilBERT reports *worse* scores in all GLUE tasks and reduces memory by only 40%. There are plenty other
orthogonal methods to save memory, such as reversible layers. We plan to discuss them, as suggested by [R2].

**Theorem 1:** We would like to thank [R3] for detailed feedback. We will use the proposed name. [R3]: *"line 88 (supp.*
*material) you forgot a minus sign."* Correct, that is a typo: the last three equations are **max**, (see Lemma 1, Eq. 3).
[R3] *"What is $q_D$, $q_{CD}$ and $q_C$ in lines 87-88 of the appendix?"* Indeed our notation needs explanation. $q_D$ denotes the
softmax denominator for query $q$, i.e. $\sum_{k \in \mathcal{K}} e^{q \cdot k}$. Similarly, $q_C$ (which we mistyped as $q_{CD}$ before L88) denotes the
softmax denominator of the cluster for query $q$, i.e. $\sum_{k \in C_t^L} e^{q \cdot k}$. We will update these. [R3]: *"It is not clear that the*
*proof works for $\epsilon=0$"*. Our argument does not require $\epsilon = 0$ or any infinities actually: There exist finite $a, \epsilon$ that suffice
(they depend on the input vectors Q, K). We only need $q_C$ to be sufficiently close to $q_D$ so that the maximizers of the
two problems are the same. For any given input instance this is a finite difference. We will rewrite our proof to avoid
infinities and have a cleaner argument. Finally, [R3] points out connections to biclustering and co-clustering. We will
discuss this and cite the relevant work, thank you for pointing this interesting connection to classical work.

**Other** [R3]: *"Does your architecture support causal masking?"* Yes. If a token gets clustered with tokens from the
future we just zero these entries in the softmax. When a token is clustered only with tokens from the future, we only
allow this token to attend to itself. [R3]: *"the paper is not clear about how $L$ is chosen."* For $O(N \log N)$ complexity,
$L$ should be $O(N)$, (mentioned in L173). In practice, we try to minimize the number of queries per cluster. Choosing
$O(1)$ queries per cluster, brings $L$ to $O(N)$. We state the number of queries per cluster in almost all our experiments
(see column $C$ of Tables 1, 2, 3, 5) to show how different choices of $L$ impact performance. We will follow the advice
of [R3] and explicitly discuss this in the Complexity Analysis section. [R4]: *"It's more insightful to show that the*
*proposed method can also work well in the pre-training setting."* We agree with [R4]. That is why we included in the
paper pre-training results for BigGAN (see Table 5). Our budget did not allow additional pre-training experiments for
NLP. [R4]: *"It's unclear how GPU friendly the method is."* Our solution is as GPU friendly as Reformer's attention,
since the codebase only differs in the LSH scheme. Our method is more useful in terms of speed for large sequences
(see Fig. 6 of our supp and Fig. 5 of Reformer). We will report GPU hours as requested by [R4]. [R4]: *"More analysis*
*would help to understand the limitations."* Please refer to "Things that did not work section". Finally, all typos and
figure suggestions will be addressed as suggested.

[Meta-Review · NeurIPS 2020]

This paper proposes a method for reducing the quadratic bottleneck of transformer architectures to O(N log N), using an asymmetric LHS clustering strategy. The paper also shows that finding an optimal assignment is NP-hard and thus, heuristic approaches must be pursued. They propose a novel type of balanced clustering algorithm to approximate attention. The method can be directly used for pre-trained models and achieves competitive/better performance with BigGAN/BERT/RoBERTa by shrinking 50% memory. There was some disagreement among reviewers about this paper, with R1 and R3 recommending solid acceptance, and R2 and R4 recommending weak reject. The reviewers mentioned as strengths that the proposed model can save memory for both training and inference; the experiments are significant compared to the memory usage of BigGAN and BERT; the proposed model can be widely applied for both CV and NLP tasks; the evaluation has shown good results on image&text data in multiple regimes: with/without fine-tuning, training from scratch, several compression levels; the analysis of the number of queries per cluster in the proposed model is quite interesting; and that the drop-in nature makes proposed method much more useful than alternatives. The main weakness is that the experiments do not compare against related efficient attention models (reformer, routing transformer, longformer) even in settings where the comparison would be possible. The comparison with reformer, in particular, seems important, given that it is also a LSH-based approach. Overall, I think this is a good paper that provides a new solution for an important problem (efficient attention is highly relevant and sought after at the moment) and validates empirically the proposed approach in different regimes. The author response addressed the lack of comparison against other methods that was pointed out as the main weakness by the reviewers, and this has been acknowledged in the discussion phase. Therefore I recommend acceptance. I urge the authors to add these new results to the final version of their paper. I also recommend them to add the citations recommended by R3. Finally, a point was made by R2 about comparison with distillation. In the discussion phase, R2 clarified that “as shown in Table 4 of TinyBERT (https://arxiv.org/pdf/1909.10351.pdf), a distilled 4-layer Transformer is comparable to the proposed method, and the distilled 6-layer model can be significantly better. But the author response makes sense because the proposed modification is not mutually exclusive with knowledge distillation.”